# Dysmenorrhea Among Women Living in Saudi Arabia: Prevalence, Determinants, and Impact

**DOI:** 10.3390/life15010108

**Published:** 2025-01-16

**Authors:** Mohammad A. Jareebi, Saja A. Almraysi, Dhiyaa A. H. Otayf, Ghadeer A. Alneel, Areej H. Zughaibi, Sarah J. Mobarki, Imtenan A. Oberi, Hayam A. Alzahrani, Shatha A. Aldhowaihi, Zainab A. Alramadhan, Majed A. Ryani, Ahmed A. Bahri, Nuha H. Abutalib, Nada M. Makein, Ahmad Y. Alqassim

**Affiliations:** 1Family and Community Medicine Department, Faculty of Medicine, Jazan University, Jazan 45142, Saudi Arabia; mjareebi@jazanu.edu.sa (M.A.J.); majedryani@gmail.com (M.A.R.); dr.bahri2010@gmail.com (A.A.B.); nabutaleb@jazanu.edu.sa (N.H.A.); nmakein@jazanu.edu.sa (N.M.M.); 2Faculty of Medicine, Jazan University, Jazan 45142, Saudi Arabia; sajaalasiri1@gmail.com (S.A.A.); dhiyaaot@gmail.com (D.A.H.O.); sarahmobarki40@gmail.com (S.J.M.); emtenan257@gmail.com (I.A.O.); 3Department of Obstetrics and Gynecology, Women’s Health Hospital, Riyadh 12231, Saudi Arabia; ghadeer-ahmad-n@hotmail.com; 4Department of Obstetrics and Gynecology, Saudi Ministry of Health, Riyadh 11992, Saudi Arabia; dr.a.z0202@hotmail.com; 5Faculty of Medicine, Batterjee Medical College, Jeddah 21442, Saudi Arabia; alzhrani7701@gmail.com; 6Faculty of Medicine, Majmmah University, Majmmah 15341, Saudi Arabia; shathaaldhowaihi@gmail.com; 7Medical Intern at King Fahad University Hospital, Bashar Ibn Burd St, Al Aqrabiyah, Al Khobar 34445, Saudi Arabia; drzainaab@gmail.com

**Keywords:** dysmenorrhea, pain, women’s health, menstrual symptoms, Saudi females, prevalence, determinants

## Abstract

Dysmenorrhea, or painful menstruation, is one of the most prevalent conditions among adolescents and young females globally, significantly affecting academic performance, quality of life, and social interactions. Despite its high prevalence, dysmenorrhea has been rarely investigated in Saudi Arabia, resulting in a scarcity of national data. This study aimed to evaluate the prevalence, severity, and determinants of dysmenorrhea among females in Saudi Arabia. This cross-sectional, questionnaire-based study included 1026 participants recruited from various regions of Saudi Arabia using a convenience sampling technique. Data collection was conducted using a validated, self-administered online questionnaire. Descriptive and inferential statistics were utilized to evaluate the prevalence, severity, and associated symptoms of dysmenorrhea. Additionally, multiple logistic regression was employed to identify the determinants of dysmenorrhea within the sample. The analysis was performed using R software. The prevalence of dysmenorrhea among the sample was 87%, with a mean pain score of 6 ± 2.2. Common menstrual cycle-associated symptoms included mood swings (79%), abdominal bloating (60%), diarrhea (32%), and constipation (26%). Factors associated with an increased risk of dysmenorrhea included Saudi nationality (OR = 1.96, *p* = 0.032), employment (OR = 1.75, *p* = 0.034), and a history of gynecological surgeries (OR = 1.81, *p* = 0.045). Conversely, multiparity was associated with a reduced risk of dysmenorrhea (OR = 0.36, *p* = 0.046). Dysmenorrhea is highly prevalent among Saudi women and is accompanied by significant menstrual symptoms that adversely affect quality of life. Understanding its determinants and associated symptoms is essential for improving management strategies and mitigating its impact on women’s lives.

## 1. Introduction

Dysmenorrhea, characterized by painful menstruation, is one of the most prevalent gynecological conditions affecting adolescent and young adult females worldwide. The condition is classified into two types: primary and secondary dysmenorrhea. Primary dysmenorrhea is characterized by idiopathic pain unrelated to pelvic pathology, typically beginning with menarche. Its diagnosis requires the exclusion of underlying pathological conditions. In contrast, secondary dysmenorrhea can be attributed to specific underlying pathologies, such as endometriosis and uterine fibroids [1,2].

The clinical presentation differs between primary and secondary dysmenorrhea. Primary dysmenorrhea typically presents with consistent pain patterns across menstrual cycles. Secondary dysmenorrhea, however, may be characterized by increasing pain severity, persistence until the end of menstruation, and unilateral presentation. Associated symptoms include gastrointestinal disturbances (nausea, vomiting, diarrhea), poor appetite, headache, and muscle cramps. Severe cases may lead to sleep disturbances, chronic pain development, and significant impairment of quality of life [1,3]. Multiple risk factors have been identified for dysmenorrhea, including younger age, smoking, abnormal body mass index (BMI), psychological conditions (depression and anxiety), menstrual irregularities (prolonged cycles, heavy bleeding), early menarche, nulliparity, and family history [4,5].

Dysmenorrhea is one of the most reported complaints among young females. A review by Ju et al. that included 15 studies found that the prevalence of dysmenorrhea among reproductive-age females were between 16% to 91%. The review also found that around 30% of the sample had reported severe pain [5]. Similarly, a systematic review by Armour et al. in 2019 found the pooled prevalence of dysmenorrhea across 38 studies was 71.1%. These findings were almost the same among secondary and university students [6]. In Saudi Arabia, a cross-sectional study was conducted in 2022, revealing that approximately 92% and 7% of the Saudi females suffered from primary and secondary dysmenorrhea, respectively [7].

Dysmenorrhea can be so severe that it negatively impacts the individual’s academic and overall quality of life. For instance, Armour et al. found that 20.1% of patients with dysmenorrhea reported absences from educational institutions, and 40.9% reported a lack of concentration in the classroom, eventually negatively impacting their performance and overall academic achievement [6]. Additionally, female workers with dysmenorrhea miss work more frequently than other female workers. Furthermore, severe dysmenorrhea adversely affects social life, contributing to the deterioration of personal relationships and the development of poor social interactions and may eventually diminish the quality of life (QoL). For example, patients with dysmenorrhea reported lower physical and psychological QoL scores compared to those with chronic diseases, such as cystic fibrosis [8,9].

Despite the global prevalence of dysmenorrhea, it has not been well investigated in Saudi Arabia, since most studies focus on certain regions. The lack of nationwide studies highlights the need for additional studies that more accurately represent the large area of Saudi Arabia, with a population around 8 million females across 13 regions. To address this, our study examined the prevalence of dysmenorrhea, patient characteristics, condition severity, and determinants among females in Saudi Arabia.

## 2. Materials and Methods

### 2.1. Study Design and Participants

This cross-sectional, questionnaire-based study was conducted in Saudi Arabia from April 2024 to November 2024. The study included adult females aged 18 to 60 residing in the Kingdom of Saudi Arabia. Females younger than 18 or older than 60, as well as those unable or unwilling to consent, were excluded. The target population comprised 7,374,595 females aged 18 to 60 years, according to the latest census [10]. A confidence interval of 95%, a margin of error of 4.5%, and a 25% non-response rate were used to calculate the minimum required sample size. A sample of 1026 was collected based on the above parameters employing a non-random convenience sampling technique. These measures were implemented to minimize sampling bias, and the calculation was verified using the following formula:$$n=\frac{Z^{2}\cdot p\cdot (1-p)}{E^{2}}$$
where *Z* represents the statistic for the desired confidence level (1.96 for 95% confidence), *p* is the expected prevalence or response distribution (0.5 used for the most conservative estimate), and *E* is the margin of error.

### 2.2. Data Collection Tool

A self-administered questionnaire was developed following a comprehensive literature review and consultation with medical specialists [1,7,11,12,13]. The questionnaire comprised three sections: sociodemographic characteristics, menstrual cycle patterns, and menstrual cycle-associated symptoms.

The sociodemographic data included questions about the participants’ age, nationality, residence, weight, height, marital status, number of children, monthly income, education, occupation, smoking habits, exercise routine, hemoglobinopathies, and history of gynecological surgery. Income categories were defined based on monthly household income in Saudi riyals (SAR) and converted to USD for international comparison. These categories reflect different socioeconomic levels in the Saudi context.

The second section focused on menstrual cycle patterns and characteristics, with questions about the age of menarche, menstrual cycle regularity, average cycle length, bleeding duration, and bleeding amount to obtain relevant information about the menstrual cycle.

The third section, termed menstrual cycle-associated symptoms, inquired about the presence of pain during menstruation, pain severity (visual analogue scale from 0–10 was used to assess pain, with scores 1–3 indicating mild pain, 4–6 indicating moderate pain, and 7–10 indicating severe pain), emergency room (ER) visits, and other associated symptoms such as abdominal bloating, mood swings, diarrhea, and constipation.

The questionnaire was piloted with 25 participants to assess its clarity, after which minor modifications were made before it was finalized. The questionnaire was distributed through various social media platforms using an online survey format, such as WhatsApp, Telegram, X (Twitter), Facebook, and other viral platforms. An explanation of the research purpose and objectives was provided to the participants, and consent was obtained before participation. Additionally, privacy and confidentiality were maintained throughout the study process.

### 2.3. Data Analysis

Following data collection, raw data were exported to Excel for error checking, validation, and assessment of missing values. Statistical analysis, including descriptive analysis and multiple logistic regression, was then performed using R software (version 4.2.3, R Foundation for Statistical Computing, Vienna, Austria) [14]. Descriptive analysis involved the use of mean, percentages, and standard deviation (SD), depending on the specific variable’s requirements. Multiple logistic regression was conducted to assess the association between dysmenorrhea and several determinants. A *p*-value of <0.05 was considered statistically significant for all tests.

### 2.4. Ethical Approval

Ethical review and approval for this study were granted by the Standing Committee for Scientific Research at Jazan University (reference number REC-45/11/1115, dated 28 April 2024). A thorough explanation of the study objectives, possible outcomes, and participants’ rights regarding voluntary withdrawal, privacy, and confidentiality was provided before obtaining consent to participate. To ensure privacy and confidentiality, no personal identifiers were collected, and data access was restricted to the research team. Additionally, this cross-sectional study was designed and reported in accordance with the Declaration of Helsinki guidelines for human subjects’ research and the STROBE (Strengthening the Reporting of Observational Studies in Epidemiology) guidelines to ensure transparency and completeness.

## 3. Results

### 3.1. Sociodemographic Characteristics

This study included 1026 participants with a mean age of 28 ± 9.5 years and a mean body mass index (BMI) of 24 ± 5.8 kg/m^2^. The majority of participants were Saudis (93%) and mostly resided in urban settings (81%), while the rest (19%) lived in rural settings. In terms of education, more than half of the participants had a bachelor’s degree (53%), while the remaining 12% and 35% had earned postgraduate studies degrees, high school degrees, or had achieved lower levels of education, respectively. As for employment, 34% of our sample comprised employed individuals, 47% were students, and 19% were unemployed. Regarding marital status, non-married/engaged individuals comprised more than half of the sample (54%), while married and divorced/widowed participants comprised 41% and 5% of the sample, respectively. The distribution of children among participants showed that 60% of participants had no children, 10% had only one child, 11% had two children, and 18% had three or more children. The distribution of monthly income indicated that 59% earned less than SAR 5000, 19% earned between SAR 5000 to 9999, 11% earned between SAR 10,000 to 14,999, and 11% earned more than SAR 15,000. Further details are provided in Table 1.

### 3.2. Habitual and Health-Related Characteristics of Study Participants

Health-related factors and lifestyle habits varied among study participants. In terms of smoking and smoking mode, only 5% of the participants were smokers, and only 1% of them smoked cigarettes, while 5% used different modes of smoking (shisha/hookah/waterpipe/vape). As for physical activity, more than half of the participants (66%) were not physically active, and only one-third of them (34%) exercised regularly. In terms of past medical and surgical history, 11% of the participants had sickle cell anemia, 7% had thalassemia, and only 16% of them had undergone a gynecological surgical intervention. Detailed information is presented in Table 2.

### 3.3. Menstrual Cycle-Related Variables’ Characteristics

Analysis of menstrual cycle-related variables revealed that the mean menarche age for the participants was 12 ± 2 years, the mean cycle length was 27 ± 7.4 days, and the mean bleeding duration was 5.2 ± 2.2 days. As for the amount of period blood, 28%, 61%, and 11% of the participating women described their bleeding as “heavy”, “moderate”, and “light”, respectively. Of the total participants, 889 (87%) reported experiencing dysmenorrhea, while 13% did not, as depicted in Figure 1. Table 3 shows more specific information about the menstrual cycle-related variables.

### 3.4. Menstrual Cycle-Associated Symptoms

Assessment of dysmenorrhea severity was conducted using multiple parameters to assess the severity of pain and the possible menstrual cycle-associated symptoms, as shown in detail in Table 4. Firstly, the participants were asked to rate their pain on a scale from 0 to 10. The highest pain intensity (10) was reported by 53 participants, accounting for approximately 5.2% of the sample. The lowest pain intensity (1), on the other hand, was reported by 45 participants, representing about 4.4% of the sample. The most frequently reported pain intensity score was 7, with 202 participants (20.2%) experiencing this level of pain. Overall, the mean pain score was 6 ± 2.2 among study participants. Figure 2 provides a graphical illustration of pain levels for further understanding. Regarding frequency, 22% of the participants experienced pain infrequently (once every 6 months), while the other 29% and 49% experienced pain occasionally (once every 3 months) and every cycle, respectively. In addition, 14% of participants had to visit the emergency department to manage the pain. As for the associated symptoms, 60% of patients suffered abdominal bloating, 79% experienced mood swings, 32% had diarrhea, and 26% suffered from constipation.

### 3.5. Determinants of Dysmenorrhea Risk

Continuing our investigation, we conducted multiple logistic regressions to identify the nature of the relationship between dysmenorrhea, sociodemographic factors, and menstrual cycle-associated variables and symptoms. In the multiple logistic regression analysis, the resulting data showed several different factors that increased the likelihood of developing dysmenorrhea. For instance, Saudi nationality was associated with a higher risk of dysmenorrhea compared to being non-Saudi, with an odds ratio (OR) of 1.96 (95% CI: 1.04–3.57, *p* = 0.032). Employment status was another factor that increased the risk of dysmenorrhea, with employed individuals (OR = 1.75, 95% CI: 1.02–3.13, *p* = 0.034) and students (OR = 1.85, 95% CI: 1.20–3.44, *p* = 0.023) facing an increased risk compared to unemployed individuals. In addition, past gynecological surgeries were also associated with a higher risk of dysmenorrhea (OR = 1.81, 95% CI: 1.02–3.43, *p* = 0.045). On the other hand, the analysis also identified factors that decreased the risk of developing dysmenorrhea, such as multiparity. Specifically, having two or more children was linked to a reduced risk of dysmenorrhea, with ORs of 0.36 (95% CI: 0.12–0.94, *p* = 0.046) and 0.36 (95% CI: 0.12–0.91, *p* = 0.042), respectively. The rest of the associations are shown in Table 5.

## 4. Discussion

This cross-sectional study aimed to evaluate the prevalence, severity, associated symptoms, and contributing factors of dysmenorrhea among women in Saudi Arabia. With a sample of 1026 participants, predominantly Saudi women (93%), this study highlights the significant burden of dysmenorrhea as a prevalent and impactful condition affecting women’s health.

The prevalence of dysmenorrhea in this study (87%) aligns with previous Saudi research, which reports rates ranging from 60.9% to 92.3% [7,14,15,16,17,18]. Similar prevalence patterns are observed globally, though with broader ranges than those reported in local studies [5]. These results suggest that dysmenorrhea is a global health issue that transcends cultural, geographical, and economic boundaries. The observed variations could partly be attributed to methodological differences. However, the consistently high prevalence underscores the necessity for standardized research methods to better assess the magnitude of the problem and to understand the true scope of this condition.

In this study, pain severity varied widely, with 20% of participants reporting severe pain (score ≥ 7 on a scale of 0 to 10). Similar trends have been observed in other Saudi studies, such as those conducted at King Abdulaziz University (38.6% experiencing severe pain) and Jeddah University (46.1% experiencing severe pain) [14,19]. A study from Al Jouf University reported that 34% of participants experienced severe pain, while others reported mild (8.7%) or moderate pain (57.3%) [20].

This finding of severe pain in 20% of participants is particularly significant, as recent evidence suggests that severe dysmenorrhea (VAS score ≥ 7) should be considered an important alarm sign for secondary dysmenorrhea, particularly endometriosis. According to a comprehensive review, endometriosis represents the main cause of secondary dysmenorrhea in adolescents and young women, with a prevalence of ultrasound signs of endometriosis reaching 35.3% in those with severe dysmenorrhea [21]. A recent study further demonstrated that among young women with dysmenorrhea and chronic pelvic pain, the percentage of endometriosis detected by ultrasound increased significantly when severe pain was present [22]. The presence of severe dysmenorrhea, especially when combined with other symptoms like dyspareunia, gastrointestinal symptoms, or heavy menstrual bleeding, should prompt a thorough evaluation to rule out secondary causes.

International studies demonstrate notable variations in reported pain severity levels. For example, in France, only 8.9% of participants described their dysmenorrhea pain as severe [23]. These variations may be attributed to cultural differences in pain perception and reporting practices, varying access to healthcare resources, or differences in the criteria and tools used to measure pain [24]. In societies where menstrual pain is stigmatized, women may underreport their symptoms, while in others with greater awareness and advocacy for women’s health, reporting may be more accurate [25].

The frequency of dysmenorrhea among participants varied significantly. Nearly half (49%) reported experiencing it with every menstrual cycle, 29% occasionally, and 22% infrequently. This distribution is consistent with findings from Jeddah University, where 66% reported experiencing dysmenorrhea during every menstrual period, although occasional and infrequent episodes were less common [19]. Such variation underscores the individual nature of dysmenorrhea and the need for personalized approaches to its management.

The study identified multiple associated symptoms that substantially contribute to the overall burden of dysmenorrhea. Commonly reported symptoms in this study included gastrointestinal disturbances such as nausea (45.7%), bloating (43.1%), and appetite changes (53.8%). These findings align with those of previous Saudi studies [16,20] and global research identifying irritability, anxiety, abdominal distension, and heightened emotional sensitivity as common symptoms of primary dysmenorrhea [26]. This symptomatology highlights the interconnected nature of menstrual health, with systemic, physical, and emotional effects contributing to the condition’s burden.

Analysis revealed multiple sociodemographic and clinical factors significantly associated with dysmenorrhea risk. In this study, Saudi nationality and employment status were associated with increased risk, reflecting potential cultural and lifestyle influences. This finding aligns with several studies that have documented higher rates of dysmenorrhea among employed women, potentially due to increased work-related stress and sedentary behavior [27]. Our finding regarding Saudi nationality contrasts with international studies that show distinct racial/ethnic patterns in disease prevalence. For instance, studies in the United States have found similar prevalence rates between White and Black populations but significantly lower rates among Hispanic and Asian groups [28]. These variations have been attributed to complex interactions between genetic, environmental, and socioeconomic factors [29,30]. This disparity might be attributed to cultural differences in pain perception and reporting, healthcare-seeking behaviors, and varying levels of awareness about menstrual health among different populations [31,32,33].

Conversely, multiparity emerged as a protective factor, consistent with findings from previous studies in Saudi Arabia and beyond. Research conducted in Taif identified similar protective effects, and studies among Zimbabwean university students also confirmed that the risk of dysmenorrhea decreases with increasing parity [17,34]. The physiological changes associated with pregnancy and childbirth, such as altered uterine contractility and hormonal regulation, may explain this protective effect [35].

Additionally, previous research has identified younger age, early menarche, and family history as significant risk factors for dysmenorrhea [17]. Although these factors were not explicitly explored in this study, their consistent presence in the literature suggests potential genetic and developmental influences on dysmenorrhea susceptibility. These findings warrant further investigation into the interplay between genetic, hormonal, and environmental factors in the pathophysiology of dysmenorrhea.

The substantial impact of dysmenorrhea on daily life is evidenced by the 14% of participants reporting recurrent emergency department visits due to severe menstrual pain. Beyond clinical encounters, dysmenorrhea also disrupts daily activities, including work, education, and social interactions. In Saudi Arabia, up to 87% of women report impairments in concentration, social activities, domestic tasks, and university attendance during menstrual periods [14,20]. Globally, studies reveal that 38% of women are unable to perform daily activities, 65.5% leave work or school early, and 68% avoid social interactions due to menstrual symptoms [36,37]. This disruption underscores dysmenorrhea’s significant impact not only on individual quality of life but also on societal productivity and healthcare systems. The inability of women to participate fully in daily life due to menstrual pain represents an unmet public health need, highlighting the importance of addressing dysmenorrhea as both a medical and social issue.

This study’s findings have important implications for both clinical practice and public health policy. First, the high prevalence and severity of dysmenorrhea necessitate increased awareness and education about menstrual health among healthcare providers, patients, and the general population. Early identification and appropriate management of dysmenorrhea can help mitigate its impact on women’s lives. Second, the risk factors identified in this study, such as employment status and gynecological surgery history, provide opportunities for targeted interventions. Workplace accommodations for menstruating women, such as flexible work schedules or access to pain management resources, could help reduce the burden of dysmenorrhea. Additionally, pre- and post-surgical counseling about potential menstrual health implications should be prioritized. The observed protective effect of multiparity suggests potential pathways for preventive interventions. While family planning decisions are deeply personal and influenced by multiple factors, healthcare providers should consider incorporating discussions about dysmenorrhea risk reduction into broader reproductive health education. Finally, the variation in symptoms and severity across different populations highlights the need for standardized diagnostic criteria and assessment tools for dysmenorrhea. Such standardization would improve the comparability of studies and facilitate the development of evidence-based guidelines for managing dysmenorrhea.

### Strengths and Limitations

Our study’s strengths include its large sample size and diverse population of Saudi females. The comprehensive assessment of prevalence, pain characteristics, associated symptoms, and risk factors provides a holistic understanding of dysmenorrhea in the Saudi context.

However, certain limitations should be acknowledged. The cross-sectional design limits causal inferences, and self-reported data may be subject to recall bias. Future longitudinal studies could provide more robust evidence regarding the temporal relationships between risk factors and dysmenorrhea. Additionally, while our study focused on a broad range of factors, there may be other variables not included in our analysis that could influence the occurrence and severity of dysmenorrhea.

## 5. Conclusions

In conclusion, this study provides invaluable insights into the prevalence and determinants of dysmenorrhea among Saudi females. The findings emphasize the necessity for targeted interventions, improved pain management strategies, and increased awareness to address this significant women’s health issue in Saudi Arabia. Future research should focus on developing and evaluating interventions tailored to the specific needs and cultural context of Saudi women, with the aim of reducing the burden of dysmenorrhea and improving overall menstrual health.

## Figures and Tables

**Figure 1 life-15-00108-f001:**
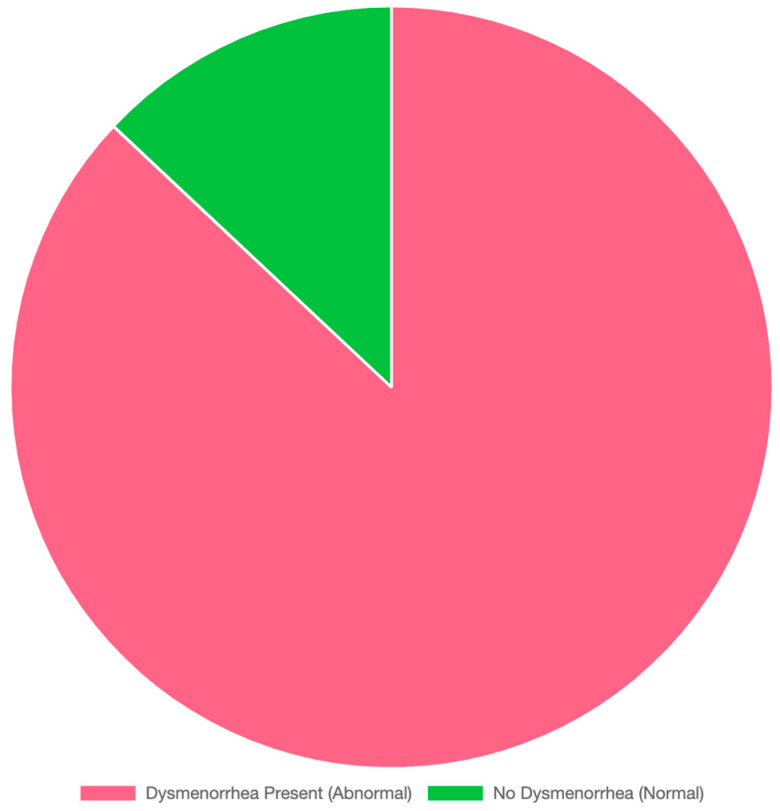
Prevalence of dysmenorrhea among the participants.

**Figure 2 life-15-00108-f002:**
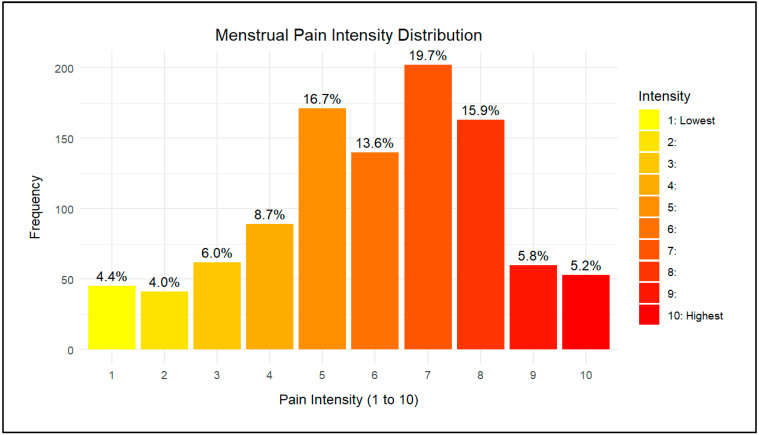
Menstrual pain intensity among participants.

**Table 1 life-15-00108-t001:** Sociodemographic characteristics of study participants (*n* = 1026).

Characteristics	Mean ± SD
Age	28 ± 9.5 years
BMI	24 ± 5.8 kg/m^2^
**Characteristics**	**Frequency (%)**
Nationality	
Saudi	950 (93%)
Non-Saudi	76 (7%)
Education	
High School Degree or Lower	362 (35%)
Bachelor’s degree	545 (53%)
Postgraduate Studies	119 (12%)
Employment	
Employed	347 (34%)
Student	479 (47%)
Unemployed	200 (19%)
Marital status	
Non-Married/Engaged	556 (54%)
Married	422 (41%)
Divorced/Widowed	48 (5%)
Number of children	
Zero	619 (60%)
One	102 (10%)
Two	117 (11%)
Three or more	188 (18%)
Income *	
“Less than 5000”	608 (59%)
“5000–9999”	193 (19%)
“10,000–14,999”	114 (11%)
“≥15,000”	111 (11%)
Residence	
Rural	200 (19%)
Urban	826 (81%)

Abbreviations: SD: standard deviation; *n*: sample size; BMI: body mass index. * Income categories: low income (<SAR 5000, <USD 1333); middle income (SAR 5000–9999, USD 1333–2666); high income (SAR 10,000–14,999, USD 2667–4000); very high income (≥SAR 15,000, ≥USD 4000). Currency conversion rate: USD 1 = SAR 3.75.

**Table 2 life-15-00108-t002:** Habitual and health-related characteristics of study participants (*n* = 1026).

Characteristics	Frequency (%)
Smoking	
Yes	56 (5%)
No	970 (95%)
Mode of smoking	
Cigarettes	8 (1%)
Shisha/Hookah/Waterpipe/Vape	50 (5%)
Exercise	
Yes	346 (34%)
No	680 (66%)
Blood disorder	
No	846 (82%)
Sickle cell anemia	111 (11%)
Thalassemia	69 (7%)
Past gynecological surgery	
Yes	167 (16%)
No	859 (84%)

Abbreviations: *n*: sample size.

**Table 3 life-15-00108-t003:** Menstrual cycle-related variables (*n* = 1026).

Characteristics	Mean ± SD
Menarche age	12 ± 2 years
Cycle length	27 ± 7.4 days
Bleeding duration	5.2 ± 2.2 days
**Characteristics**	**Frequency (%)**
Blood amount	
Heavy (more than 1 fully soaked sanitary pad every 2 h)	111(11%)
Moderate (more than 1 soaked sanitary pad within 3 h)	629 (61%)
Light (less than 1 soaked sanitary pad in 3 h)	286 (28%)
Dysmenorrhea	
Yes	889 (87%)
No	137 (13%)

Abbreviations: SD, standard deviation; *n*, sample size.

**Table 4 life-15-00108-t004:** Menstrual cycle-associated symptoms (*n* = 1026).

Characteristics	Mean ± SD
Pain score (out of 10)	6 ± 2.2
**Characteristics**	**Frequency (%)**
Pain frequency	
In a few cycles (once every 6 months)	222 (22%)
In some cycles (once every 3 months)	297 (29%)
In every cycle	507 (49%)
Emergency department visits	
Yes	144 (14%)
No	882 (86%)
Abdominal bloating	
Yes	619 (60%)
No	407 (40%)
Mood swings	
Yes	810 (79%)
No	216 (21%)
Diarrhea	
Yes	324 (32%)
No	702 (68%)
Constipation	
Yes	270 (26%)
No	756 (74%)

Abbreviations: SD, standard deviation; *n*, sample size.

**Table 5 life-15-00108-t005:** Multiple Logistic Regression of study variables and Dysmenorrhea risk.

Predictors	Dysmenorrhea		
	OR	95% CI	*p*
Age	1.00	0.96–1.03	0.773
Nationality (reference: non-Saudi)			
[Saudi]	1.96	1.04–3.57	**0.032**
Marital status (reference: single)			
[Married]	1.65	0.67–4.58	0.308
[Divorced/Widowed]	2.19	0.67–8.72	0.229
Income (reference: less than SAR 5000) †			
[Between SAR 5000–9999]	0.87	0.49–1.58	0.642
[Between SAR 10,000–15,000]	0.76	0.38–1.56	0.438
[>SAR 15,000]	0.48	0.22–1.05	0.062
Education (reference: high school or lower)			
[Bachelor]	0.99	0.62–1.55	0.959
[Postgraduate]	1.27	0.60–2.77	0.541
Employment (reference: unemployed)			
[Employed]	1.75	1.02–3.13	**0.034**
[Student]	1.85	1.20–3.44	**0.023**
Number of children (reference: no children)			
[One child]	0.50	0.17–1.36	0.182
[Two children]	0.36	0.12–0.94	**0.046**
[Three children or more]	0.36	0.12–0.91	**0.042**
Residence (reference: rural)			
[Urban]	1.18	0.72–1.88	0.493
Smoking status (reference: no)			
[Yes]	0.61	0.31–1.30	0.174
Past gynecological surgeries (reference: no)			
[Yes]	1.81	1.02–3.43	**0.045**
Menarche Age	0.96	0.87–1.06	0.372
BMI	1.01	0.97–1.05	0.689
Observations	1026		

† Income categories are defined as follows: Low income (<SAR 5000, <USD 1333); middle income (SAR 5000–9999, USD 1333–2666); high income (SAR 10,000–14,999, USD 2667–4000); very high income (≥SAR 15,000, ≥USD 4000). Currency conversion rate: USD 1 = SAR 3.75.

## Data Availability

The data sets produced and examined in the course of this study can be obtained from the corresponding author, provided the request is reasonable.

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
