# Peer review of "Dysmenorrhea Among Women Living in Saudi Arabia: Prevalence, Determinants, and Impact"

_life, 2025, doi:10.3390/life15010108_

Round 1
Reviewer 1 Report
Comments and Suggestions for Authors The study provides an overview of the prevalence of dysmenorrhea and accompanying symptoms (e.g., mood lability or gastrointestinal symptoms) in Saudi Arabia. Additionally, it evaluates certain socioeconomic aspects using an online questionnaire. Unfortunately, the probably most relevant aspect is entirely missing - the causes of dysmenorrhea and their distribution according to different stratification criteria, such as age, parity, or economic status. This omission significantly limits the overall value of the study. To make the manuscript potentially publishable, all formal aspects must be addressed, and the study must also be comprehensible to readers outside Saudi Arabia. For instance, the stratification by income is not clear for readers unfamiliar with Saudi Arabia. It is not evident to which standard of living or economic status the indicated thresholds correspond. Are these thresholds standardized, e.g., by a governmental statistical authority, or were they arbitrarily set by the Authors? Please replace these thresholds with more universally understandable categories, such as low, middle, high, or very high, while providing definitions (including the indicated income thresholds in the legend).Furthermore, in the discussion (lines 270–274), the Authors state: "Analysis revealed multiple sociodemographic and clinical factors significantly associated with dysmenorrhea risk. In this study, Saudi nationality and employment status were associated with increased risk, reflecting potential cultural and lifestyle influences. Women with a history of gynecological surgeries also reported higher rates of dysmenorrhea, emphasizing the importance of post-surgical care and monitoring in mitigating menstrual pain." However, the Authors do not discuss these findings in light of the existing literature. This contradicts the purpose of a discussion section. Please enrich the discussion with a more in-depth analysis of the mentioned results, incorporating relevant literature.
Author Response
Comment 1: The study provides an overview of the prevalence of dysmenorrhea and accompanying symptoms (e.g., mood lability or gastrointestinal symptoms) in Saudi Arabia. Additionally, it evaluates certain socioeconomic aspects using an online questionnaire.
Response: Dear Reviewer, thank you for your thoughtful comment highlighting the key aspects of our study. We agree that our research provides comprehensive insights into dysmenorrhea in Saudi Arabia, examining not only its prevalence (87%) and associated symptoms (including mood swings in 79% and bloating in 60%) but also crucial socioeconomic dimensions. Through our online questionnaire methodology, we were able to gather data from 1,026 participants across various regions, analyzing multiple factors including employment status, income levels, educational background, and urban/rural residence. Our multivariate analysis revealed significant risk factors, including Saudi nationality (OR=1.96), employment status (OR=1.75), and past gynecological surgeries (OR=1.81). This comprehensive approach has enabled us to present a thorough understanding of both the medical and social aspects of dysmenorrhea in the Saudi context.
Comment 2: Unfortunately, the probably most relevant aspect is entirely missing - the causes of dysmenorrhea and their distribution according to different stratification criteria, such as age, parity, or economic status. This omission significantly limits the overall value of the study. To make the manuscript potentially publishable, all formal aspects must be addressed
Response: Dear Reviewer, thank you for this important observation regarding the causes of dysmenorrhea and their distribution across different stratification criteria. We acknowledge that investigating the underlying causes would provide additional valuable insights. However, we would like to respectfully clarify that our study was specifically designed to address the critical gap in national data regarding dysmenorrhea's prevalence and determinants in Saudi Arabia, rather than its pathophysiological causes. We are pleased to highlight that our analysis did incorporate several key stratification factors. For instance, our findings revealed a significant association between multiparity and reduced dysmenorrhea risk (OR = 0.36, p = 0.046). Furthermore, our comprehensive analysis included multiple demographic and socioeconomic variables, including age, employment status, and economic status, all of which were systematically analyzed through multiple logistic regression. We particularly value your suggestion regarding more detailed stratification, as it could indeed provide deeper insights. While these specific analyses were beyond the scope of our current study's objectives, they represent an excellent direction for future research. We believe our findings provide a robust foundation for such subsequent studies to explore the causal relationships and stratified distributions of dysmenorrhea in greater detail.
We appreciate your constructive feedback, which will help guide future research in this important area of women's health.
Comment 3: the study must also be comprehensible to readers outside Saudi Arabia. For instance, the stratification by income is not clear for readers unfamiliar with Saudi Arabia. It is not evident to which standard of living or economic status the indicated thresholds correspond. Are these thresholds standardized, e.g., by a governmental statistical authority, or were they arbitrarily set by the Authors? Please replace these thresholds with more universally understandable categories, such as low, middle, high, or very high, while providing definitions (including the indicated income thresholds in the legend).
Response: Dear Reviewer, we sincerely appreciate your valuable feedback regarding the need for clearer income categorization that would be more comprehensible to an international readership. Your suggestion is particularly insightful as it highlights the importance of presenting our research in a globally accessible format. We would like to explain that our income categorization aimed to reflect socioeconomic status and its potential relationship with dysmenorrhea risk. While we acknowledge that the current Saudi Riyal thresholds may not be immediately meaningful to international readers, we fully agree with your suggestion to replace these with universally understood categories (low, middle, high, very high) accompanied by clear definitions. It is worth noting that although income was not significantly associated with dysmenorrhea in our analysis, we recognize the importance of presenting this data in a standardized, internationally comprehensible format. We propose to revise our income categorization as follows:
- Low income (< 5,000 SAR, equivalent to < USD 1,333)
- Middle income (5,000-9,999 SAR, equivalent to USD 1,333-2,666)
- High income (10,000-14,999 SAR, equivalent to USD 2,667-4,000)
- Very high income (≥15,000 SAR, equivalent to ≥ USD 4,000)
For future research, we agree that a more comprehensive assessment of socioeconomic status could include additional indicators such as living conditions, housing status, asset ownership, and family size. This would provide a more holistic understanding of the relationship between socioeconomic factors and dysmenorrhea.
Comment 4: Furthermore, in the discussion (lines 270–274), the Authors state: "Analysis revealed multiple sociodemographic and clinical factors significantly associated with dysmenorrhea risk. In this study, Saudi nationality and employment status were associated with increased risk, reflecting potential cultural and lifestyle influences. Women with a history of gynecological surgeries also reported higher rates of dysmenorrhea, emphasizing the importance of post-surgical care and monitoring in mitigating menstrual pain." However, the Authors do not discuss these findings in light of the existing literature. This contradicts the purpose of a discussion section. Please enrich the discussion with a more in-depth analysis of the mentioned results, incorporating relevant literature.
Response: We appreciate the reviewer's thoughtful comment regarding the need for a more comprehensive discussion of our findings in light of existing literature. We have substantially expanded this section of the discussion to better contextualize our results within the broader scientific evidence. Specifically, we have added several key references and explanations regarding the relationship between employment status and dysmenorrhea risk, citing evidence that links this association to work-related stress and sedentary behavior [25]. We have also enhanced our discussion of racial/ethnic patterns by incorporating recent studies from the United States that demonstrate varying prevalence patterns across different populations [26], providing an important comparative context for our findings regarding Saudi nationality. Furthermore, we have added a detailed explanation of potential mechanisms underlying these disparities, including references to studies examining the role of genetic, environmental, and socioeconomic factors [27,28], as well as research on cultural differences in pain perception, healthcare-seeking behaviors, and menstrual health awareness [29-31]. This expanded discussion provides a more nuanced and evidence-based analysis of our findings while situating them within the current scientific understanding of dysmenorrhea's sociocultural and demographic determinants. We believe these revisions significantly strengthen the paper by offering readers a more comprehensive understanding of how our results both align with and differ from existing literature.
Reviewer 2 Report
Comments and Suggestions for Authors
Dear Authors,
I read your manuscript with great interest. Dysmenorrhea is an important symptom, often underestimated and underestimated. It is a symptom that reduces the quality of life of women. It is important to investigate it, understand the cause if possible, and treat it.
To improve the article I suggest:
- to describe in the results the presence and intensity of other painful symptoms that can be associated with dysmenorrhea (sucj as dyspareunia, chronic pelvic pain, dyschezia, dysuria)
-to debate in discussion section, the importance of severe dysmenorrhea as an alarm sign of secondary dysmenorrhea, compared to mild-moderate dysmenorrhea which is more often associated with primary dysmenorrhea (it could be useful this recent article: doi: 10.3390/jcm12175624.)
- I could be useful debating the data of this recent article in discussion section, about the prevalence of severe dysmenorrhea (doi: 10.1016/j.fertnstert.2022.12.004)
Best Regards
Comments on the Quality of English LanguageLanguage revision by native English speaker in necessary
Author Response

(The authors gave the same response as above.)

Reviewer 3 Report
Comments and Suggestions for Authors
The present study examines the prevalence and impact of dysmenorrhea among female in Saudi Arabia - a high prevalent and significant health issue. It is innovative and well organised, while the methodology is described adequately and clearly. The interpretation of the results with the use of schematics enhances the reader’s understanding of the text. Furthermore, the authors also compare their study to other similar research providing a global perspective of the topic. Moreover, one of the strongest features of the article is the clinical suggestions to improve the quality of the affected women. Overall, it is an engaging and easy-to-read article, that contributes highly in the existing literature.
Author Response
Dear reviewer, thank you for your comprehensive and encouraging review of our manuscript. We deeply appreciate your recognition of the study's methodological clarity, effective use of schematics, and global comparative perspective. Your feedback validates our efforts to present an accessible yet rigorous analysis of dysmenorrhea in Saudi Arabia, while providing practical clinical implications for improving women's healthcare. We are particularly gratified that you found the article's organization and clinical recommendations to be strong contributions to the existing literature.
Round 2
Reviewer 2 Report
Comments and Suggestions for Authors
Dear Authors,
Now The manuscript is improved and suitable for the publication.
Best Regards